

# An optimized approach to germ-free rearing in the jewel wasp *Nasonia*

J. Dylan Shropshire[1,*], Edward J. van Opstal[1,*] and Seth R. Bordenstein[1,2]

[1] Biological Sciences, Vanderbilt University, Nashville, TN, United States
[2] Pathology, Microbiology & Immunology, Vanderbilt University, Nashville, TN, United States
[*] These authors contributed equally to this work.

## ABSTRACT

Development of a *Nasonia in vitro* germ-free rearing system in 2012 enabled investigation of *Nasonia*-microbiota interactions and real-time visualization of parasitoid metamorphosis. However, the use of antibiotics, bleach, and fetal bovine serum introduced artifacts relative to conventional rearing of *Nasonia*. Here, we optimize the germ-free rearing procedure by using filter sterilization *in lieu* of antibiotics and by removing residual bleach and fetal bovine serum. Comparison of these methods reveals no influence on larval survival or growth, and a 52% improvement in adult production. Additionally, adult males produced in the new germ-free system are similar in size to conventionally reared males. Experimental implications of these changes are discussed.

## INTRODUCTION

The *Nasonia* genus (*Ashmead & Smith, 1904*) consists of four closely related interfertile parasitoid wasp species and has been a powerful model for the study of genetics (*Davies & Tauber, 2015*; *Lynch, 2015*; *Raychoudhury et al., 2010*), evolution (*Bordenstein, O'Hara & Werren, 2001*; *Bordenstein & Werren, 2007*; *Brucker & Bordenstein, 2013*; *Clark et al., 2010*), endosymbiosis (*Bordenstein, O'Hara & Werren, 2001*; *Ferree et al., 2008*), development (*Rivers & Losinger, 2014*; *Verhulst et al., 2013*; *Zwier et al., 2012*), behavior (*Baeder & King, 2004*; *Beukeboom & Van den Assem, 2001*; *Clark et al., 2010*; *Drapeau & Werren, 1999*; *Raychoudhury et al., 2010*), pheromonal communication (*Diao et al., 2016*; *Ruther & Hammerl, 2014*; *Steiner, Hermann & Ruther, 2006*), and other areas. The design and publication of an *in vitro* system for *Nasonia* in 2012 detached *Nasonia* from its fly host, allowed for real-time monitoring of development, and provided an avenue to study how microbes influence *Nasonia* biology (*Brucker & Bordenstein, 2012a*). These tools advanced the *Nasonia* system to explore how gut microbiota influence development and hybrid lethality (*Brucker & Bordenstein, 2013*).

*Nasonia* germ-free rearing involves two major components: (i) sterilizing *Nasonia* embryos and (ii) providing larvae with sterilized food in an *in vitro* system. Embryo sterilization is conducted by picking *Nasonia* embryos from pupal fly hosts (typically *Sarcophaga bullata*; *Werren & Loehlin, 2009a*) and then rinsing the embryos with bleach followed by sterile water (*Brucker & Bordenstein, 2012a*). Producing *Nasonia* Rearing Medium (NRM)

Corresponding authors
J. Dylan Shropshire,
dylan.shropshire@vanderbilt.edu
Seth R. Bordenstein,
s.bordenstein@vanderbilt.edu

involves the collection of hundreds of fly pupae, extraction of proteinaceous fluids from those pupae, addition of fetal bovine serum (FBS) and Schneider's *Drosophila* medium for additional nutrition, filter sterilization, and addition of antibiotics (Fig. 1; *Brucker & Bordenstein, 2012a*). Sterilized embryos are then placed on a transwell permeable membrane with filter-sterilized NRM underneath for feeding (*Brucker & Bordenstein, 2012a*).

This protocol yielded similarly sized *Nasonia* to those from *in vivo* rearing (*Brucker & Bordenstein, 2012a*). However, NRM production relies on introducing foreign and potentially harmful elements such as bleach, FBS, and antibiotics. Removal of each component carries its own rationale. For example, the bleach treatment was intended to kill surface bacteria on the puparium of host flies and remove particulates (*Brucker & Bordenstein, 2012a*). However, surface bacteria will be removed during filtration and residual bleach from the rinse may persist in the final NRM product as a toxic agent. Furthermore, FBS is added as a nutritional supplement to increase larval survival and development (*Brucker & Bordenstein, 2012a*), but *Nasonia* do not frequently encounter components of FBS including bovine-derived hormones such as testosterone, progesterone, insulin, and growth hormones (*Honn, Singly & Chavin, 1975*). Finally, antibiotics are a confounding variable and removing them will provide more flexibility to bacterial inoculations in the *in vitro* system.

This study removes these three major components of the original NRM and optimizes the procedure by eliminating extraneous steps and utilizing quicker approaches. These changes are validated by directly comparing germ-free *Nasonia* reared on either the original (NRMv1) or optimized (NRMv2) media for larval and pupal survival, larval growth, and adult production. The morphology of adults produced both *in vitro* and *in vivo* is then compared.

## MATERIALS AND METHODS

### *Nasonia* rearing medium (NRMv1)

*Sarcophaga bullata* pupae were produced as previously described (*Werren & Loehlin, 2009a*). Approximately 150 ml of *S. bullata* pupae were transferred to a sterile 250 ml beaker after close inspection to remove larvae, poor quality pupae, and debris. A solution of 10% Clorox bleach was then added to the beaker to cover the pupae. After five minutes, the bleach was drained from the beaker and the pupae were repeatedly rinsed with sterile millipore water until the scent of bleach was absent. Sterile millipore water was added in the beaker to approximately 2/3 the volume of pupae, covered, and placed in a 36 °C water bath to soften the puparium. *S. bullata* pupae were homogenized using a household kitchen blender and filtered through a 100 µm cell strainer (Fisherbrand; Thermo Fisher scientific Incorporated, Waltham, MA, USA). The filtrate was poured evenly across two 50 ml conical tubes (Falcon, Corning Incorporated, Corning, NY, USA) and centrifuged at 4 °C (25,000xG) for 5 min to separate the sediment, protein, and lipid layers, and a 22 gauge needle (BD PrecisionGlide; Becton, Dickinson and Company, Franklin Lakes, NJ, USA) was used to remove the protein layer. The protein layer was combined with 50 ml of Schneider's *Drosophila* medium 1 x and 20% FBS. Using a reusable 500 ml vacuum filtration apparatus (Nalgene, Thermo Fisher scientific Incorporated, Waltham, MA, USA), the resulting product was passed through filter paper (Whatman; General Electric Healthcare Life Sciences, Maidstone, United Kingdom)

A       B

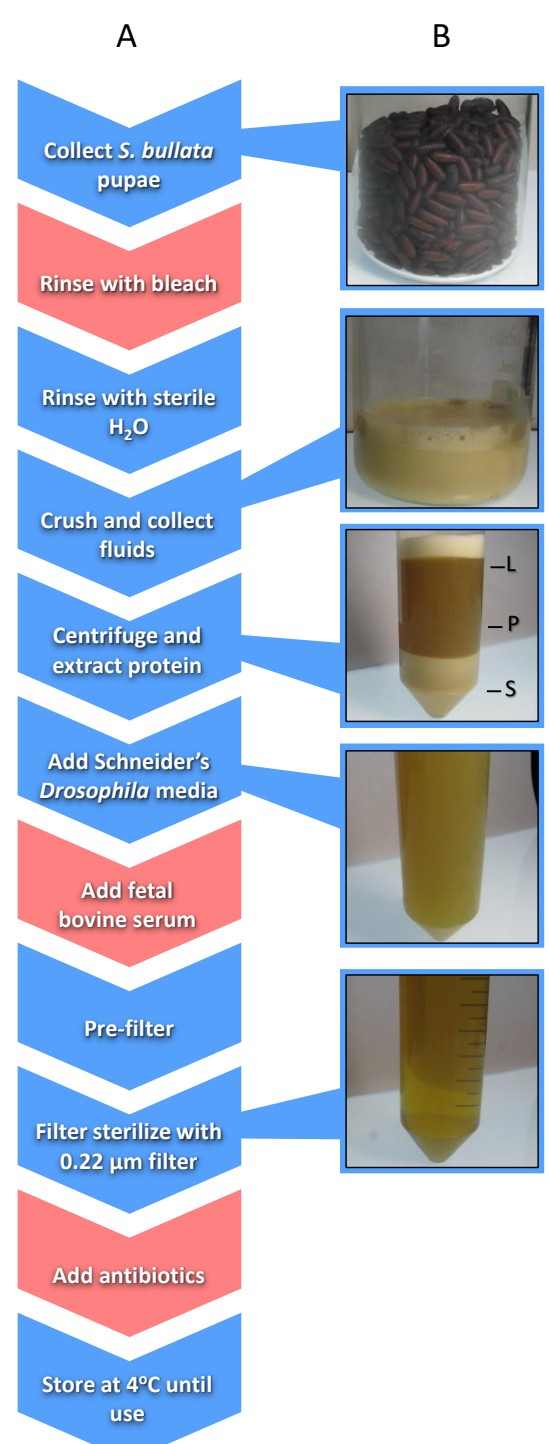

**Collect *S. bullata* pupae**

**Rinse with bleach**

**Rinse with sterile H$_2$O**

**Crush and collect fluids**

**Centrifuge and extract protein**

**Add Schneider's *Drosophila* media**

**Add fetal bovine serum**

**Pre-filter**

**Filter sterilize with 0.22 µm filter**

**Add antibiotics**

**Store at 4ºC until use**

**Figure 1   Schematic of the workflow to produce *Nasonia* Rearing Media (NRM).** (A) Red boxes indicate steps present in NRMv1 but eliminated in NRMv2; blue boxes indicate steps present in both procedures. (B) shows the visual progression from *S. bullata* pupae to final NRM product. L, lipid layer; P, protein layer; S, sediment layer.

with gradually smaller pore sizes (11, 6, 2.5, 0.8, and 0.45 μm). A 0.22 μm syringe filter (Costar, Corning Incorporated, Corning, NY) was used to remove bacteria. Finally, 200 μg of carbenicillin and penicillin/streptomycin were added to the medium. The final product was stored at 4 °C until use (Fig. 1).

### *Nasonia* rearing medium (NRMv2)

Approximately the same number of *S. bullata* pupae were collected as described above. Pupae were subsequently rinsed in sterile millipore water to remove small particulates. They were then crushed by hand through a 100 μm nylon net (EMD Millipore, Merck Millipore, Billerica, MA, USA) and the filtrate was collected in a sterile 250 ml glass beaker. Nylon powder-free non-sterile gloves were worn during this extraction. The filtrate was centrifuged at 4 °C (25,000xG) for 10 min to separate the sediment, protein, and lipid layers. Using a 22 gauge needle (BD PrecisionGlide; Becton, Dickinson and Company, Franklin Lakes, NJ), the protein layer was transferred to a sterile beaker. Schneider's *Drosophila* media was added to the protein extract to triple the volume and, using a reusable vacuum filtration apparatus (Nalgene; Thermo Fisher scientific Incorporated, Waltham, MA, USA), the resulting mixture was passed through filters with gradually smaller pore sizes (11, 6, 2.5, 0.8, and 0.45 μm). A 0.22 μm syringe filter (Costar; Corning Incorporated, Corning, NY, USA) was used to remove bacteria. The final product was stored at 4 °C until use (Fig. 1). The following is a step-wise protocol for the production of NRMv2.

1. Fill a sterilized beaker with 150 ml of *S. bullata* pupa. Remove larvae, poor quality pupae, and debris.
2. In the beaker, cover pupae with sterile Millipore water, allow to sit for 1 min, and strain to remove surface particulates from the puparium surface. Some moisture will remain on the pupae.
3. Crush the pupae by hand (covered with powder-free nitrile gloves) and squeeze juices through a 100 μm nylon mesh to remove the *S. bullata* puparium.
4. Separate juices (approximately 70–90 ml) evenly into two 50 ml conical tubes and seal tightly.
5. Centrifuge the mixture for 10 min at 4 °C (25,000xG). The mixture will separate into three distinct layers: a sediment, protein, and lipid layer from bottom to top, respectively.
6. To prevent clogging during filtration, extract the protein layer using a 22 gauge sterile needle and transfer it to a sterile beaker under sterile laminar flow.
7. Add a 2:1 ratio of Schneider's *Drosophila* medium to the protein extract.
8. Using a vacuum filtration system, filter the media through progressively smaller pore sizes (11, 6, 2.5, 0.8, and 0.45 μm filters) to remove increasingly smaller particulates. To prevent clogging, replace filter paper when flow begins to slow.
9. Sterilize the media by filtering through a 0.22 μm syringe filter, taking care to use aseptic technique.
10. Store at 4 °C for up to 2 weeks.
11. Filter NRM through a 0.22 μm syringe filter before use to ensure sterility and remove sedimentation.
### *Nasonia* strains and collections

*N. vitripennis* (strain AsymCx; *Wolbachia* uninfected) mated females were hosted on *S. bullata* pupae and housed in glass culture tubes capped with cotton at $25 \pm 2$ °C in constant light, as previously described (*Werren & Loehlin, 2009b*). After 10–12 days, *S. bullata* puplariums were opened and virgin *N. vitripennis* females were collected as pupae from the resulting offspring. Upon adult eclosion, individual virgin females were isolated and provided two *S. bullata* pupae for hosting to increase the number of eggs deposited in subsequent hostings. In haplodiploids, virgin females are fecund and lay all male (haploid) offspring. Two days after initial hostings, females were provided with a new *S. bullata* pupae housed in a Styrofoam plug, allowing her to oviposit only on the anterior end of the host for easy embryo collection.

### Germ-free rearing of *Nasonia*

*N. vitripennis* strain AsymCx embryos were extracted from *S. bullata* pupae parasitized by virgin females after 12–24 h. 20–25 embryos were placed on a 3 µm pore transwell polyester membrane (Costar; Corning Incorporated, Corning, NY, USA) and sterilized twice with 70 µl 10% bleach solution and once with 70 µl 70% ethanol solution. The embryos were then rinsed three times with 80 µl sterile millipore water. After rinsing, the transwell insert was moved into a 24 well plate with 250 µl of NRM in the well. All plates were stored in a sterile Tupperware box at $25 \pm 2$ °C in constant light conditions for the duration of the experiment. Under sterile laminar flow, transwells were moved to new wells with 250 µl of fresh NRM every second day. Approximately 1.5 ml of NRM was used per transwell over the duration of the experiment. After eleven days, the transwells were moved to dry wells in a clean plate and the 12 empty surrounding wells were filled with 1 ml of sterile millipore water to increase humidity. Two plates with 12 transwells each (total of 24) were set up using either NRMv1 or NRMv2 by JDS for *Nasonia in vitro* rearing in May 2016. Replicate rearing and collection of larval and pupal survival data was conducted on both NRMv1 ($N = 9$ inserts) and NRMv2 ($N = 13$) by EVO in April 2015 and March 2016 respectively (Fig. S1).

### Comparative analysis of development

A picture was taken of each well, every day for 20 days, under magnification using a microscope-attached AmScope MT1000 camera. A baseline for the number of larvae present in a well was determined by counting the number of larvae present in transwell pictures three days after embryo deposition on the transwell membranes (day 3). Survival estimates were determined by counting the number of live larvae on day 6 and the number of live larvae and pupae on day 14, compared to day 3. Larvae and pupae were identified as dead if they were visibly desiccated or malformed. Larval length was determined using ImageJ software by measuring the anterior to posterior end of larvae on days 3, 6, and 14. The proportion of adults produced by a transwell was determined as follows: (the number of larvae on day 3 − the number of dead larvae and pupae remaining on day 20) ÷ the number of larvae on day 3. Pictures of conventionally reared and germ-free (NRMv2) adult males were taken, and ImageJ was used to measure head width, which is a correlate for body size in *Nasonia* (*Blaul & Ruther, 2012*; *Tsai et al., 2014*).

## RESULTS

Larval growth of *Nasonia vitripennis* reared on NRMv1 was previously compared to conventionally reared *N. vitripennis* and there were no differences in larval survival or larval growth over development (*Brucker & Bordenstein, 2012a*). Here we demonstrate, in comparisons between NRMv1 and NRMv2, that there is also no difference in larval and pupal survival to day 14 (Fig. 2A; Mann–Whitney U (MWU) for day 6 $p = 0.19$ and day 14 $p = 0.41$) nor length, measured as the distance from the anterior to posterior end (Fig. 2B; MWU for day 3 $p = 0.26$, day 6 $p = 0.18$, day 14 $p = 0.13$). A replicate experiment reveals that the survival results are repeatable (Fig. S1; MWU for day 6 $p = 0.23$ and day 14 $p = 0.06$). Moreover, a visual comparison of larval sizes on NRMv1 and NRMv2 shows no major differences (Figs. 2C–2F). These findings indicate that removal of residual bleach, FBS, and antibiotics does not have a significant impact on larval survival or development.

NRMv1 yielded low adult survival compared to conventional rearing (*Brucker & Bordenstein, 2012a*). To investigate if using NRMv2 improves larval to adult survival, both the number of transwells producing adults and the average number of adults produced per transwell were compared between NRMv1 and NRMv2. The number of transwells that produced at least a single adult did not differ between NRMv1 (79% productive; $N = 24$) and NRMv2 (88% productive; $N = 24$; Fig. 3A; Fisher's exact test $p = 0.7$). However and importantly, NRMv2 yielded a higher proportion of adults than NRMv1 (Fig. 3B; MWU $p = 0.001$), accounting for a 52% increase in larval to adult survival. Finally, to ensure that adults produced in the *in vitro* system are similar in size to conventional adults, the head width of adult males produced on NRMv2 ($N = 16$) was compared to conventionally reared ($N = 16$) adult males, and there was no significant difference (Fig. 3C; MWU $p = 0.72$).

## DISCUSSION

The previously established *Nasonia in vitro* germ-free rearing protocol (*Brucker & Bordenstein, 2012a*), which involved sterilizing embryos and feeding the larvae NRMv1, was crucial for conducting experiments on *Nasonia*-microbiota interactions (*Brucker & Bordenstein, 2013*). However, this initial version of the germ-free rearing system contained highly artificial elements such as bleach rinsing, FBS, and antibiotics (Fig. 1; NRMv2). Following removal of these elements, we show that the alterations to the NRM did not influence larval and pupal survival to day 14 in replicate experiments (Fig. 2A; Fig. S1) or larval growth (Fig. 2B), but importantly resulted in a 52% increase in larval to adult survival (Fig. 3B). Moreover, the size of adult males produced on NRMv2 and *in vivo* do not differ (Fig. 3C), suggesting that both *in vitro* and *in vivo* rearing produce morphologically similar adults.

Aside from making the *Nasonia in vitro* system more biologically relevant, the new media has multiple experimental implications. For example, antibiotics are a confounding variable with unknown consequences to *Nasonia* biology, and they can hinder inoculation capabilities of the system by causing bacterial communities introduced to rapidly shift in composition. Thus, removal of antibiotics in NRMv2 makes it easier to derive conclusions and may provide more flexibility for inoculations *in vitro*, namely introduction of full microbial communities derived from *Nasonia* species. This new system permits the introduction of

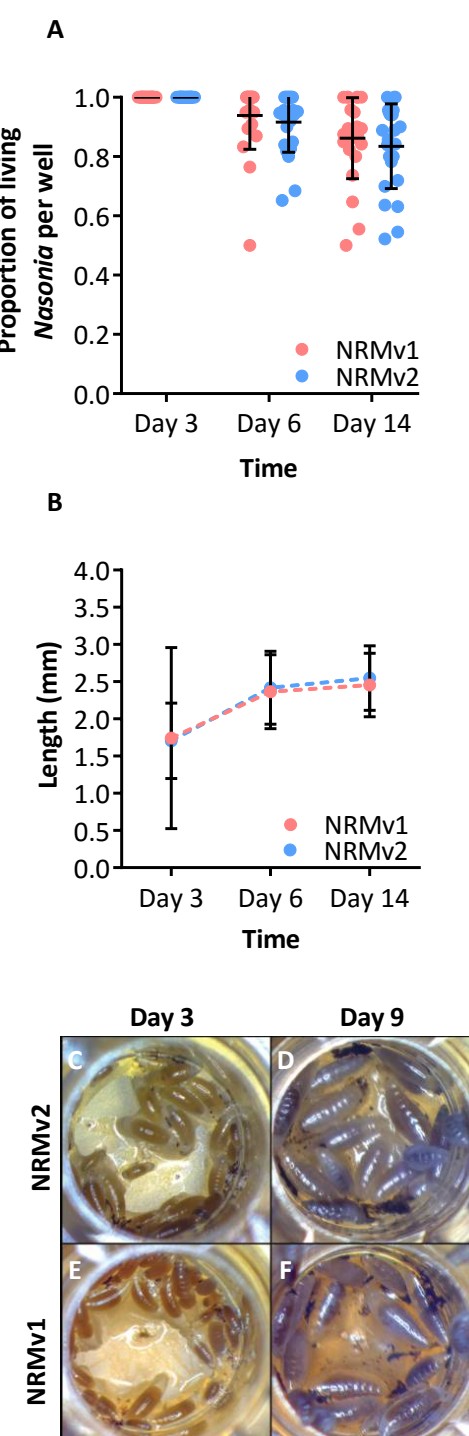

**Figure 2 Comparison of *Nasonia* germ-free larval development on NRMv1 and NRMv2.** (A) The number of living *Nasonia vitripennis* in transwells on days 3, 6, and 14. There are no statistically significant differences in larval survival on NRMv1 and NRMv2. (B) Equivalent larval lengths measured from anterior to posterior end in mm. (C–F) Visual comparison of larvae reared on NRMv1 and NRMv2 on days 6 and 9. Vertical bars with caps represent standard deviation from the mean.

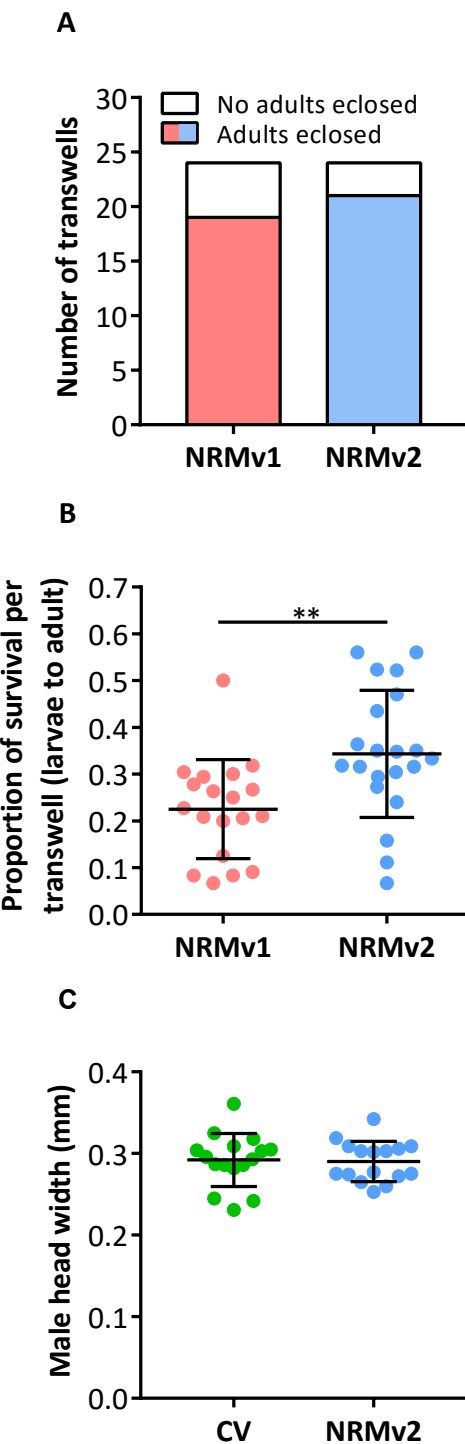

**Figure 3 Survival and size of *Nasonia* germ-free adult males.** (A) Comparison of the number of transwells producing adults between NRMv1 and NRMv2. (B) Proportion of larval to adult survival in each transwell is determined as follows: (the number of larvae on day 3 − the number of larvae and pupae remaining on day 20) ÷ the number of larvae on day 3. (C) Adult head widths from germ-free males reared on NRMv2 and males reared conventionally. Larval to adult adult survival was statistically different between the two media (Mann–Whitney U, *P*-value = 0.001). All other measures were not significant with $\alpha = 0.05$. Vertical bars with caps represent standard deviation from the mean.

both autochthonous and allochthonous microbial communities, enabling investigations of the functional relevance of host-specific microbial communities or microbial species. For example, the *Nasonia* microbiota exhibits "phylosymbiosis," a pattern in which microbial community relationships parallel the phylogenetic relationships of the host species (*Brucker & Bordenstein, 2012b*; *Brucker & Bordenstein, 2013*). Transplanting communities between species will test the functional relevance of phylosymbiosis.

Furthermore, improved survival of larvae to adults on NRMv2 makes obtaining sample sizes of adults and the measurement of adult phenomes (e.g., physiology, anatomy, and behavior) more feasible. In this context, NRMv2 permits improved exploration of *Nasonia* adult-microbiota interactions. For example, there are many examples of microbe-mediated signals used in mate-choice, species recognition, and kin recognition (Reviewed in *Shropshire & Bordenstein, 2016*). *Nasonia* species produce several different signals including cuticular hydrocarbons (*Buellesbach et al., 2013*), abdominal sex pheromones (*Diao et al., 2016*), and cephalic pheromones housed in an oral gland (*Miko & Deans, 2014*; *Ruther & Hammerl, 2014*). This *in vitro* rearing system allows for the exploration of the interaction of microbes with host signals to test what role these complex interactions may have in adult behavior, insect communication, and reproductive isolation.

Parasitoid wasps are also difficult to study developmentally because the fly host's puparium obstructs visualization of the *Nasonia* larvae and pupae, preventing multiple measures of a single individual over time. *In vitro* rearing of *Nasonia* allows for observations of single individuals over developmental time and for strict control of larval diet, bacterial exposure, and *Nasonia* density. Using this system, one may test how these variables influence metamorphosis (*Johnston & Rolff, 2015*), wing and body size (*Rivers & Losinger, 2014*), craniofacial anomalies (*Werren et al., 2015*), and many other physiological traits.

In summary, we streamlined and improved upon the *Nasonia in vitro* rearing system while removing antibiotics and other factors from the equation. These changes open the door to multidisciplinary studies of host-microbiota interactions and development and add to *Nasonia*'s utility as a model system.

### Funding

This research was funded by NSF Awards DEB 1046149 and IOS 1456778 to SRB. The funders had no role in study design, data collection and analysis, decision to publish, or preparation of the manuscript.

### Grant Disclosures

The following grant information was disclosed by the authors:
NSF: DEB 1046149, IOS 1456778.

### Competing Interests

The authors declare there are no competing interests.

## Author Contributions

- J. Dylan Shropshire conceived and designed the experiments, performed the experiments, analyzed the data, wrote the paper, prepared figures and/or tables, reviewed drafts of the paper.
- Edward J. van Opstal conceived and designed the experiments, performed the experiments, analyzed the data, wrote the paper, reviewed drafts of the paper.
- Seth R. Bordenstein conceived and designed the experiments, analyzed the data, wrote the paper, prepared figures and/or tables, reviewed drafts of the paper.

## Data Availability

The raw data is included in the manuscript.

## Supplemental Information

Supplemental information for this article can be found online at http://dx.doi.org/10.7717/peerj.2316#supplemental-information.

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
