# Peer review of "An optimized approach to germ-free rearing in the jewel wasp Nasonia"

_PeerJ, doi:10.7717/peerj.2316_

## Round 0.1 · original submission · Minor Revisions

Please find attached the comments with a few suggestions. Sorry, but the letter above is a canned response that I cannot edit-the manuscript does not "require a number of Minor Revisions", only several that may help readers understand the protocol better. I look forward to the resubmission and appreciate your submission to PeerJ.

·

Basic reporting

This represents a clever improvement of a technique for ex vivo rearing of parasitoid wasps, is well-written, and is clearly within the scope of PeerJ. The review criteria here appear to be acceptable.

Experimental design

- There is a flaw in the experimental design that should be reported. Specifically, it appears that only one preparation of each type of medium was tested. Since the goal of the study is to demonstrate that the new media preparation method performs adequately, it is critical to know if similar results are attainable when the /preparation/ method is repeated. Note that the observed differential performance of the two media may reflect batch-to-batch differences rather than the effects of the protocol changes. This can only be disentangled by replication. Given that the authors’ goal is to show that the new media does not perform worse than the old media, a high degree of precision does not seem necessary here, just enough evidence shown that the method is reliable. An example of an experiment which would address this issue would be a smaller scale repeat of one of the phenotypic assays (e.g., larval to adult viability) for a few batches of both types of media.

- Please mention at the appropriate place in the Methods that a Protocol appears at the end.

- Methods & protocol section, regarding centrifuging of homogenate: The language here should be clarified. Step 4 of the protocol reads as if 70mL of homogenate was centrifuged in a 50mL conical tube, and no mention of centrifuge tube/bottle size is given in the methods.

- Filters and cell strainers - Please list part name, size, manufacturer, and any additional apparatus. Are these disposable units (e.g., Steri-Flips), sintered glass filter units, syringe filters? How was filtration applied in each case - gravity, vacuum (pump, water-flow), syringe pressure?

The other review criteria are acceptable.

Validity of the findings

See comment above about repeatability of the medium preparation method. The other review criteria are acceptable.

Additional comments

No additional comments.

Reviewer 2 ·

Basic reporting

Upon review of the manuscript “An Optimized Approach to Germ-free Rearing in the Jewel Wasp Nasonia”, I highly recommend that this manuscript be accepted. This manuscript is clear and very well written. The introduction and background are relevant and appropriately referenced. This is a well researched and prepared manuscript.

Experimental design

The experimental design is original and appropriate for the improvement of in vitro rearing of Nasonia.

Validity of the findings

This manuscript introduces a protocol for a germ-free media preparation and Nasonia rearing that facilitates further studies that were previously limited or not possible due to confounding antibiotic components and bovine hormones not found in in vivo development. This is important because these cofounders can skew developmental outcomes or microbial associated investigations. The authors address the issue of media optimization for in vitro Nasonia rearing in which they eliminate confounding attributes of the media and mitigate possible problems this would present with modified protocol steps to ensure the elimination of contaminants and an outcome more similar to in vivo development. These research finding are valid and pose benefits not only to Nasonia research but may be beneficial to other developmental research models as well.

Additional comments

You have prepared a well written manuscript that was a pleasure to read. As a reviewer, I greatly appreciate your efforts. Best to you in your future research endeavors.

Reviewer 3 ·

Basic reporting

The paper meets PeerJ's standards and shows a new and improved way to rear Nasonia. NRMv2 will be helpful to others and allows for a standardized method for the field. Also allows for less expensive and a simpler way to rear wasps without antibiotics.

Experimental design

The authors show an improved and compelling way to rear Nasonia.

Validity of the findings

The authors do a great job comparing the NRMv1 and NRMv2. I only have one simple issue- The caption in Fig 2 says (A) shows the number of living Nasonia and it should read the proportion of living Nasonia.

---

## Round 0.2 · accepted · Accept

Thank you for submitting to PeerJ, and for carefully considering and editing according to reviewer comments.